# Revisiting degron motifs in human AURKA required for its targeting by APC/C^FZR1

Ahmed Abdelbaki[1],*, Camilla Ascanelli[1],* (ID), Cynthia N Okoye[1],* (ID), H Begum Akman[1] (ID), Giacomo Janson[2], Mingwei Min[1] (ID), Chiara Marcozzi[1] (ID), Anja Hagting[1], Rhys Grant[1], Maria De Luca[1], Italia Anna Asteriti[3], Giulia Guarguaglini[3] (ID), Alessandro Paiardini[2] (ID), Catherine Lindon[1] (ID)

**Mitotic kinase Aurora A (AURKA) diverges from other kinases in its multiple active conformations that may explain its interphase roles and the limited efficacy of drugs targeting the kinase pocket. Regulation of AURKA activity by the cell is critically dependent on destruction mediated by the anaphase-promoting complex (APC/C^FZR1) during mitotic exit and G1 phase and requires an atypical N-terminal degron in AURKA called the "A-box" in addition to a reported canonical D-box degron in the C-terminus. Here, we find that the reported C-terminal D-box of AURKA does not act as a degron and instead mediates essential structural features of the protein. In living cells, the N-terminal intrinsically disordered region of AURKA containing the A-box is sufficient to confer FZR1-dependent mitotic degradation. Both in silico and in cellulo assays predict the QRVL short linear interacting motif of the A-box to be a phospho-regulated D-box. We propose that degradation of full-length AURKA also depends on an intact C-terminal domain because of critical conformational parameters permissive for both activity and mitotic degradation of AURKA.**

## Introduction

The control of mitotic exit is a paradigm for cellular regulation through targeted proteolysis. The process is orchestrated by a multi-subunit ubiquitin ligase (E3) complex, the anaphase-promoting complex (APC/C), which uses two WD40 domain co-activator paralogues contributing to substrate recognition, CDC20 and Cdh1/FZR1 (henceforth referred to as FZR1). E3s recognize their targets through substrate motifs called degrons, and although some degrons conform to a robust consensus (e.g., the di-phospho-degrons recognized by SCFβTRCP), most are rather ill-defined and can usefully be defined as short linear interacting motifs (SLiMs) present in intrinsically disordered regions (IDRs) of proteins. SLiMs are proposed to adopt transient structures to mediate weak protein–protein interactions such as those between the substrate and the E3 ligase (Van Roey et al, 2014). Such structures have now been solved by cryo-EM to identify docking sites on the APC/C for degrons known as the D-box (destruction box), KEN motif, and ABBA motif (Chao et al, 2012; He et al, 2013; Chang et al, 2015; Brown et al, 2016). The first D-box was identified in the N-terminal IDR of cyclin B1 (Glotzer et al, 1991); subsequently, a large number have been found in other APC/C substrates, most fitting the general consensus RxxLxxxxx, although a number of variants have been identified that lack either R at position 1 (P1) or L at P4 (He et al, 2013; Davey & Morgan, 2016). The KEN motif is extremely common in the proteome, first identified in CDC20 (Pfleger & Kirschner, 2000) when it was thought to be specific to substrates targeted by the FZR1-activated version of the APC/C but now revealed to be a universal APC/C degron that docks onto the upper surface of the WD40 propeller of either CDC20 or FZR1 (Chao et al, 2012; He et al, 2013). The ABBA motif has been identified in BubR1 and cyclin A and confers increased affinity that is essential for control of the mitotic checkpoint (Di Fiore et al, 2015; Qin et al, 2016).

It seems likely that a combination of degrons for multivalent docking at the KEN receptor and the D-box receptor (DBR) in the

[1]Department of Pharmacology, University of Cambridge, Cambridge, UK    [2]Department of Biochemical Sciences, Sapienza University of Rome, Rome, Italy    [3]Institute of Molecular Biology and Pathology, National Research Council of Italy, c/o Sapienza University of Rome, Rome, Italy

Correspondence: acl34@cam.ac.uk
Ahmed Abdelbaki's present address is Human Embryo and Stem Cell Laboratory, The Francis Crick Institute, London, UK
Ahmed Abdelbaki's present address is Department of Zoology, Faculty of Science, Zagazig University, Zagazig, Egypt
H Begum Akman's present address is Department of Biological Sciences, METU, Ankara, Turkey
Giacomo Janson's present address is Department of Biochemistry and Molecular Biology, Michigan State University, East Lansing, MI, USA
Mingwei Min's present address is Guangzhou Laboratory, Guangzhou, China
Chiara Marcozzi's present address is Centro Andaluz de Biología Molecular y Medicina Regenerativa CABIMER, Universidad de Sevilla-CSIC-Universidad Pablo de Olavide, Seville, Spain
Anja Hagting's present address is University of Cambridge Department of Medicine, MRC-Laboratory of Molecular Biology, Cambridge, UK
Rhys Grant's present address is Department of Biochemistry, University of Cambridge, Cambridge, UK
*Ahmed Abdelbaki, Camilla Ascanelli, and Cynthia N Okoye contributed equally to this work

APC/C is required to generate selectivity in substrate targeting amongst the large number of degron motifs present in the proteome, and to generate the increased affinity of the APC/C–substrate interaction required for efficient ubiquitination (Lu et al, 2015). As discussed in Davey & Morgan (2016), the lack of strict conservation of degron sequences can be explained by multivalency of degron–E3 interactions and participation of residues outside the consensus. These features go hand in hand with flexibility of IDRs, and flexibility in sequences surrounding the degron is essential for substrate lysines to be able to mount nucleophilic attack on a nearby ubiquitin thioester linkage. The lack of sequence conservation between degrons has contributed to historic confusion in the field, with some atypical degrons described as "novel" being subsequently redefined as variants on known degrons (i.e., they dock to the known receptor sites on the APC/C) (He et al, 2013; Davey & Morgan, 2016). Indeed, a systematic study of 16 APC/C$^{FZR1}$ substrates in budding yeast concluded that all of them depended on the DBR of FZR1 for their degradation, even when they did not have an obvious D-box (Qin et al, 2016). One candidate atypical D-box that has not been further investigated is the so-called "A-box" of AURKA (Littlepage & Ruderman, 2002).

AURKA is a substrate of the APC/C during anaphase that, unlike other anaphase substrates, is specific to the FZR1-bound form of the APC/C (Honda et al, 2000; Castro et al, 2002a; Kitajima et al, 2007; Floyd et al, 2008; Min et al, 2015). Potential degrons in AURKA have been defined mostly through in vitro studies, with deletion or mutation of either the "A-box" motif or the putative D-box shown to stabilize AURKA against mitotic degradation in one or more assays in vitro or in living cells (summarized in Table 1). There is widespread acceptance in the literature that destruction of full-length AURKA depends upon an N-terminal A-box and C-terminal D-box.

However, it is also well known from the crystal structure of the AURKA kinase domain that the putative D-box is located in a highly structured domain—inconsistent with the definition of a SLiM—and that its conserved RxxL residues are structurally buried (Bayliss et al, 2003). The proposed degron would therefore be inaccessible to APC/C binding unless ubiquitination of AURKA was preceded by an unfolding step, rendering problematic the designation of this motif as a degron, despite the in vitro evidence of degron function.

**Table 1. Summary of AURKA degrons described in the scientific literature.**

| Reported AURKA degron | | Evidence in vitro | Evidence in cells | Notes |
|---|---|---|---|---|
| **Name** | **Position** | | | |
| | Xenopus Human | | | |
| D-box[a] | R378 | Arlot-Bonnemains et al (2001) | | RxxL > RxxI stabilizes in CSF extract assay |
| | | Castro et al (2002a) | | RxxL > AxxA stabilizes and reduces FZR1-dependent ubiquitination |
| | | Littlepage & Ruderman (2002) | | 1-386 or RxxL > AxxL is the stable version |
| | R371 | Crane et al (2004) | | In vitro degradation assays |
| | | | Kitajima et al (2007) | Sensitivity to FZR1 overexpression |
| KEN[b] | K6 | Arlot-Bonnemains et al (2001); Littlepage & Ruderman (2002) | | Very small effect on degradation, not thought to be a degron |
| | K5 | Crane et al (2004) | | Very small effect on degradation, not thought to be a degron |
| | | | Min et al (2014) | Identified as ubiquitinated residue |
| A-box | 34–69; 44–55 | Littlepage & Ruderman (2002) | | 1-136 sufficient for degradation |
| | | Castro et al (2002b) | | A-box behaves as a "D-box–activating domain" |
| | 32–65 | Crane et al (2004) | | In vitro degradation assays |
| | | | Floyd et al (2008) | Live-cell mitotic degradation |
| QRVL[b] | Q45 | Crane et al (2004) | | In vitro degradation assays |
| | | | Kitajima et al (2007) | Sensitivity to FZR1 overexpression. |
| Phospho site[c] | S53 | Littlepage & Ruderman (2002); Littlepage et al (2002); Castro et al (2002b) | | Phospho-mimic versions stable in in vitro degradation assays. |
| | S51 | Crane et al (2004) | | Phospho-mimic versions stable in in vitro degradation assays. |
| | | | Kitajima et al (2007); Lindon et al (2015) | Sensitivity to FZR1 overexpression. Live-cell mitotic degradation. |

[a]Also shown for the AURKB D-box in vitro in Stewart & Fang (2005) but not in Nguyen et al (2005).
[b]Identified as degron in AURKB (Nguyen et al, 2005).
[c]pS51 found in proteomic studies as a *bona fide* phosphorylation site (Daub et al, 2008; Dulla et al, 2010; Plotnikova et al, 2010).

Here, we describe a study undertaken to provide a more complete characterization of AURKA degrons. We report that the proposed C-terminal D-box does not display properties of a degron and instead mediates structural features of the protein essential for its normal function. We show that in living cells, as has previously been reported in vitro (Littlepage & Ruderman, 2002), the N-terminal IDR of AURKA is sufficient to confer FZR1-dependent mitotic degradation and that both in silico docking experiments and live-cell degradation assays predict the so-called "A-box" to be an N-terminal D-box. Nonetheless, degradation of full-length AURKA also depends on an intact C-terminal domain, because of critical conformational parameters. We propose an explanation of how mitotic degradation of AURKA is sensitive to the conformational dynamics of the substrate.

## Results and Discussion

A number of publications identify putative degrons in AURKA through studies on both Xenopus and human proteins (Table 1). Because the existence of a D-box within the structured kinase domain of AURKA has been called into question (Lindon et al, 2015; Davey & Morgan, 2016), we decided to look more closely at the putative D-box status of this motif through a combined analysis of structure, function, and degradation of different AURKA mutants.

The structure of the hsAURKA kinase domain (122–403) was examined in PyMOL, and variations in free energy resulting from different point mutations in the putative D-box ($R_{371}$xxL) or in an adjacent D-box–like motif ($R_{375}$xxL) were calculated using the FoldX3 software (Fig 1). The molecular structure of the putative D-box shows L374 fitted into the hydrophobic aliphatic pocket on the kinase domain and a salt bridge established between the R371 and the conserved residue E299 (Fig 1A). Gibbs free energy variations ($\Delta\Delta G$) for the protein folding state predict that the RxxL > AxxA substitution frequently used to test for D-box function is strongly destabilizing to the structure (R371A/L374A, $\Delta\Delta G$ = 5.8 kcal mol$^{-1}$). R371A contributes most of the free energy variation (4.9 kcal mol$^{-1}$) with L374A less destabilizing ($\Delta\Delta G$ = 2.6 kcal mol$^{-1}$). The conserved substitution L374I is predicted to destabilize least of all ($\Delta\Delta G$ = 1.6 kcal mol$^{-1}$) (Fig 1B).

We examined the localization of YFP- or Venus-tagged versions of AURKA in living cells and found that versions with structurally destabilizing substitutions in the $R_{371}$xxL motif did not behave like the WT protein, being excluded from (R371A/L374A) or only weakly present (R371A) on the microtubules of the mitotic spindle (Fig 1C). The less destabilizing substitution (RxxI) did not appear to affect localization in mitotic cells. In contrast, the "non-degradable" version of AURKA (ΔA-box) generated by deletion of the N-terminal A-box degron (Castro et al, 2002b; Littlepage & Ruderman, 2002) has previously been shown to localize in an identical fashion to the WT protein (Floyd et al, 2008; Lindon et al, 2015).

The various subcellular localizations of the Aurora kinases are known to depend on their binding partners, with the best characterized partner of AURKA, TPX2, responsible for its localization to the mitotic spindle (Kufer et al, 2002). We therefore compared the localization of AURKA $R_{371}$xxL motif mutants alongside the known

loss of the interaction mutant AURKA S155R (Bibby et al, 2009) by quantitative immunofluorescence (IF) of fixed mitotic cells (Fig 1D), and quantified directly their interaction with TPX2 using an in situ proximity ligation assay (isPLA; Fig 1E). As expected, we found that the R371A/L374A version of AURKA, which does not localize to the mitotic spindle (Fig 1C and D), showed strongly reduced interaction with TPX2 (Figs 1E and S1). Quantification of isPLA signals confirmed that the amount of TPX2 interaction measured for different $R_{371}$xxL mutants was consistent with the amount of spindle localization, with R371A/L374A showing a more marked effect on both spindle localization and TPX2 interaction than S155R. Probing the same versions of AURKA with antibody against the phospho-T288 epitope by immunoblot confirmed that detection of the autoactivated kinase is severely compromised in R371A/L374A, as predicted for the S155R version (Fig S2). We concluded that perturbed conformation of the C-terminal region of its kinase domain prevents AURKA folding and function required for interaction with TPX2, and activation.

Next, we compared our panel of putative D-box substitutions for their effect on mitotic degradation of AURKA-Venus using a fluorescence time-lapse assay (Figs 2 and S3). The R371A/L374A version was resistant to mitotic degradation, as is the ΔA-box version lacking the previously characterized N-terminal degron sequence located between residues 32 and 66 (Fig 2A). The R371A substitution, with a partial destabilizing effect on AURKA, showed partial resistance to mitotic degradation. L374I, with the lowest $\Delta\Delta G$, had no effect on mitotic degradation of the protein (Fig 2B). Plotting the amount of degradation (as a percentage of each version of AURKA remaining 60 min after anaphase) against the calculated $\Delta\Delta G$ of folding revealed a significant inverse correlation of the two values (Fig 2C). These results suggested that a perturbed structure could be responsible for both the deficiency in interaction with TPX2 and the deficiency in mitotic degradation of AURKA $R_{371}$xxL (the so-called "D-box") mutants, and did not allow us to conclude whether or not the $R_{371}$xxL motif was a *bona fide* degron.

Therefore, we investigated further whether the different $R_{371}$xxL mutations showed a pattern of mitotic degradation consistent with the designation of this motif as a D-box. Our observations that Ala substitutions at P1 and P4 of the putative D-box stabilized AURKA against mitotic degradation were consistent with the known literature on D-box motifs. However, our finding that L > I at P4 (L374I) had no effect on mitotic degradation was not consistent. An isoleucine sidechain at P4 of a D-box would be predicted to disrupt binding to the DBR because of the shifted methyl in isoleucine that reduces the van der Waals contribution of this residue to the interaction in the hydrophobic pocket, and should be sufficient to stabilize a substrate against APC/C-mediated degradation. Indeed, this had previously been reported for mutation of the equivalent P4 residue in Xenopus Aurora A (L381I) in in vitro degradation assays (see Table 1). We tested the prediction on a well-known anaphase substrate of the APC/C, Polo-like kinase 1 (Plk1), whose degradation is critically dependent on a single D-box ($R_{337}$xxL; Fig 2D) (Lindon & Pines, 2004). We introduced the single L > I substitution at P4 of the Plk1 D-box and found that this version, L340I, was unable to mediate any mitotic degradation of correctly localized Plk1 (Fig 2E). Taken together, our findings support the prediction that the L > I substitution at P4 abrogates D-box function and that—because in our

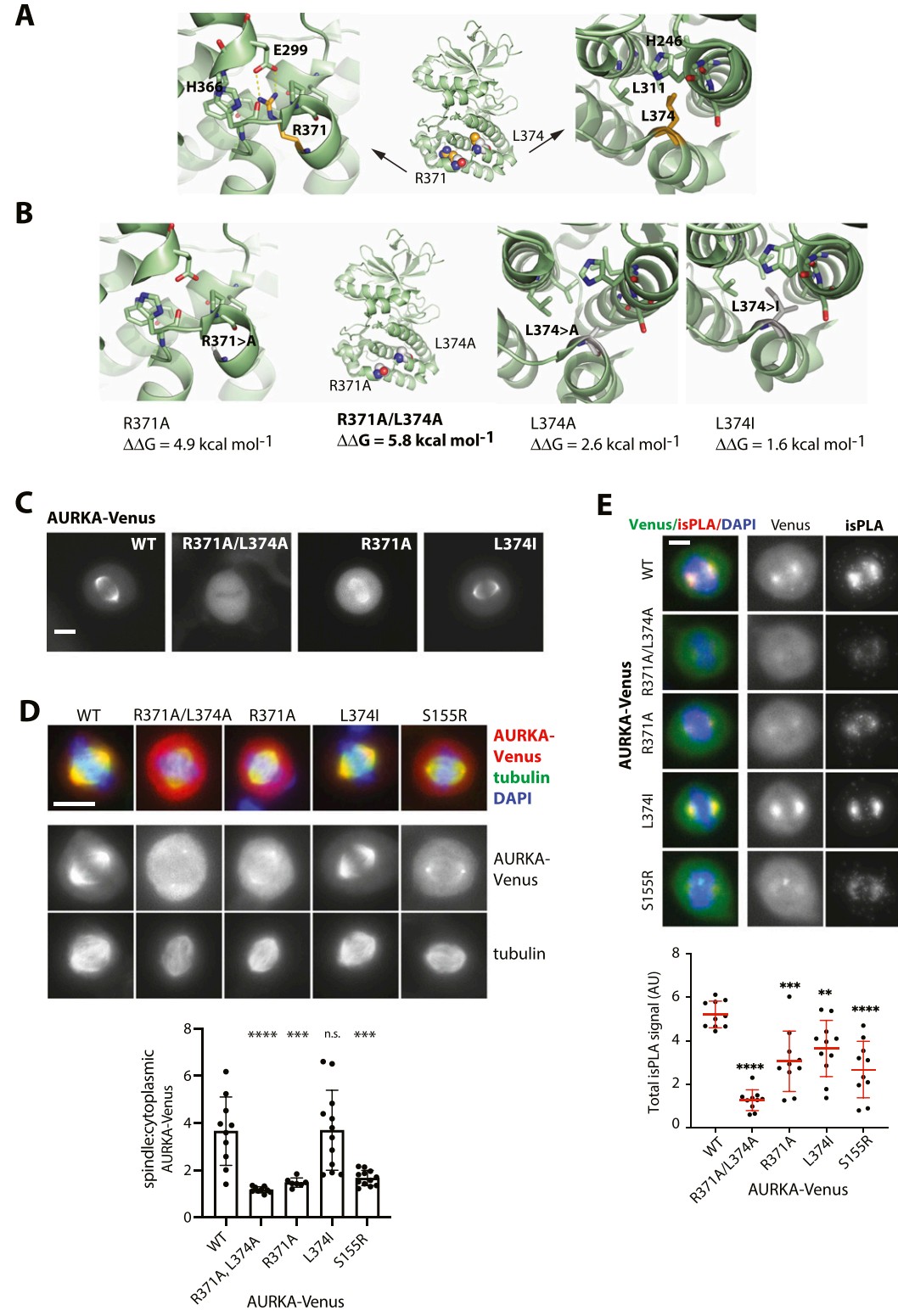

**Figure 1. C-terminal R₃₇₁xxL motif of AURKA plays a critical role in folding and function.**

**(A, B)** In silico testing of the C-terminal D-box–like motif. **(A)** Published structures of the AURKA C-terminal region (here, PDB ID:1MQ4) show the R$_{371}$xxL motif buried within the kinase domain. Arginine residue R371 (orange) establishes salt bridges with conserved glutamic acid residue E299 (green). Leucine L374 fits into the hydrophobic aliphatic pocket on the kinase domain. **(A, B)** Interactions shown in (A) are lost in predicted structures modelled for R371A and L374A substitutions. Gibbs free energy variations (ΔΔG = ΔGmut-ΔGwt) for the protein folding state were predicted using the FoldX3 software and show that R371A and L374A substitutions are more strongly destabilizing to the structure than the conserved substitution L374I. **(C, D, E)** Characterization of different versions of Venus-tagged AURKA in human U2OS cells.

experiments, AURKA L374I is degraded normally—the $R_{371}$xxL motif in human AURKA is unlikely to function as a D-box degron.

Because the C-terminal RxxL motif of AURKA was probably not a D-box, we investigated whether the N-terminal IDR would be sufficient—and necessary—for mitotic degradation in live-cell assays. The A-box (32–66) has been shown in a number of studies to be essential for AURKA degradation in a manner that depends critically on the strongly conserved $Q_{45}$RVL SLiM and on the phosphorylation status of S51 (Table 1). The N-terminal $K_5$EN motif contributes to degradation, although because K5 is a ubiquitination site during mitotic exit (Min et al, 2013), it is not clear whether this motif acts as a degron, provides the ubiquitin receptor in the substrate, or both. We first tested AURKA (1-133) fused to GFP in live-cell assays (Fig S4) and subsequently AURKA (1-67) fused to mNeonGreen (mNeon), after confirming that the mNeon-tagged version was degraded more efficiently than the GFP-tagged version (as reported in Khmelinskii et al [2016]). We found that AURKA (1-67) was sufficient to direct anaphase-specific, FZR1-dependent degradation of fluorescent protein tags (Fig 3A). The full-length protein was degraded more efficiently than the N-terminus alone, but we could conclude nonetheless that AURKA (1-67) contains degrons required for its targeted destruction at mitotic exit. We further tested, in the context of AURKA (1-67), features contributing to the specificity of AURKA targeting at mitotic exit, finding that both $K_5$EN and $Q_{45}$RVL SLiMs were required for degradation (Min et al, 2013) (Fig 3B). Therefore, AURKA (1-67) recapitulates known features of AURKA degradation, although the reduced efficiency of degradation compared with full-length protein leaves open the possibility that additional degrons exist.

We turned to an in silico docking approach to examine whether the atypical $Q_{45}$RVL degron might bind one of the known degron receptor sites on FZR1. We docked the peptide $Q_{45}$RVLCPSNS into the sites on FZR1 (D-box, KEN, and ABBA) identified from the cryo-EM structure of the FZR1 WD40 domain bound to the pseudosubstrate domain of Acm1 (PDB ID:4BH6; He et al, 2013) (Figs 3C and S5), using the FlexPepDock server (Raveh et al, 2010). The optimized pose at each site was compared with binding of a cognate degron and the statistically optimized atomic potential assigned to each interaction. This revealed that the $Q_{45}$RVL peptide favoured interaction at the D-box pocket compared with other sites and that docking of $Q_{45}$RVL at this site was energetically comparable with docking of D-box peptides (Fig 3C). Comparison of the A-box with a panel of D-boxes showed that it docks with an affinity within the range of known D-boxes, although with reduced affinity compared with more canonical D-boxes (Fig S5). We note that in silico docking to FZR1 alone ignores any contacts made between the substrate

and the APC10 that probably contribute to the affinity of substrate for the D-box binding pocket (Qin et al, 2019), and therefore, we may have underestimated the likely preference of $Q_{45}$RVL for the DBR.

The $Q_{45}$RVLCPSNS peptide can be docked in a similar pose to canonical D-box peptides, such that the peptide adopts a 90° turn at P4 with the leucine sidechain extending into the hydrophobic cleft in FZR1 identified by He et al (2013) (Fig 3D), and consistent with L at P4 being the most critical residue of the D-box (He et al, 2013; Davey & Morgan, 2016). We tested the importance of the leucine sidechain by making the L > I substitution at P4, the "D-box test" predicted to disrupt docking at the DBR (and shown to block degradation of Plk1 in Fig 2D). The L48I version of AURKA (1-67) was resistant to anaphase degradation, consistent with the hypothesis that the so-called "A-box" ($Q_{45}$RVL) is a D-box (Fig 3E). Finally, we compared directly the ability to degrade AURKA (1-67) of the newly designated D-box at Q45 with the D-box of Plk1 (at R337) and the previously designated D-box of AURKA (at R371), by substituting these motifs at the same position in AURKA (1-67). This experiment showed that although Q45 D-box mediates less efficient degradation than the Plk1 D-box, the AURKA R371 motif mediates no degradation at all, confirming that the motif at Q45 is the functional D-box in AURKA (Fig 3F).

Given the conservation of R at P1 in most D-boxes, how does the AURKA $Q_{45}$RVL dock efficiently at the DBR? In silico docking predicts that some of the electrostatic interactions with E465 and D180 of FZR1, usually made by R at P1 of canonical D-boxes, could be replaced by R at P2 (Fig 4A). In He et al (2013), it is argued that contacts made by P7 of the D-box could compensate for the lack of R at P1. In AURKA, P7 of the $Q_{45}$xxL D-box is occupied by S51, which contacts D180 in the docking pose shown in Fig 4A. We note that in alternative poses, where the P7 residue does not contact D180 we observe steric clashes between D-box residues at P8 and P9, and APC10. A block on degradation caused by the phosphomimetic substitution at S51 has been a consistently observed feature of AURKA mitotic degradation over two decades (Table 1 and Fig 4B). We confirmed in cell-based assays that S51D abrogates ubiquitination of full-length AURKA-Venus as efficiently as mutation of $Q_{45}$RVL or deletion of the entire A-box (32–66), pointing to a critical role of the residue in P7 in docking at the DBR (Fig 4C). We then compared in silico docking of WT A-box peptide (S at P7) with its "non-degradable" version (D at P7), computing free energy values using FoldX3. This analysis confirmed increased binding energy of the mutant (Fig 4D), resulting from electrostatic repulsion between the negative charge at P7 and D180 of FZR1 (Fig 4A). The ~2.0 kcal/mol difference that we observed translates into a predicted ~30-fold increase in $K_d$ and is consistent with the inhibition of mitotic

---

**(C)** Panels showing localization of AURKA-Venus in live cells during interphase and at mitosis. **(D)** U2OS cells transfected with different variants of AURKA-Venus were MeOH-fixed and processed for immunofluorescence using antibodies against GFP (red), and β-tubulin (green) and DAPI (blue). Representative images of mitotic cells are shown (upper panels), with quantified data on spindle localization of AURKA variants shown in the accompanying graph (lower panel): average pixel values within an ROI of fixed size were measured at spindle (next to but not overlapping centrosome) and in the cytoplasm (midway between spindle and cortex) and, after subtraction of background values (neighbouring the cell), used to calculate the spindle:cytoplasmic ratio. Data from individual mitotic cells (n ≥ 10) are plotted, with a bar chart indicating the mean and SDs. **(E)** Interaction with TPX2 assayed by isPLA (see also Fig S1). isPLA signal revealed AURKA-Venus–TPX2 interaction on mitotic spindles (examples in upper panels) with quantified data shown in the accompanying graph (lower panel): total isPLA signal was measured per mitotic cell, corrected for background, and normalized to the mean of the WT values in each experiment. Data for each condition (n ≥ 10) were plotted in a scatter plot with bar and whiskers to indicate the mean and SDs. **(D, E)** Each mutant was compared with WT by ordinary one-way ANOVA. **$P ≤ 0.01$; ***$P ≤ 0.001$; ****$P ≤ 0.0001$; and n.s., non-significant. Results are representative of two identical repeats of experiments.

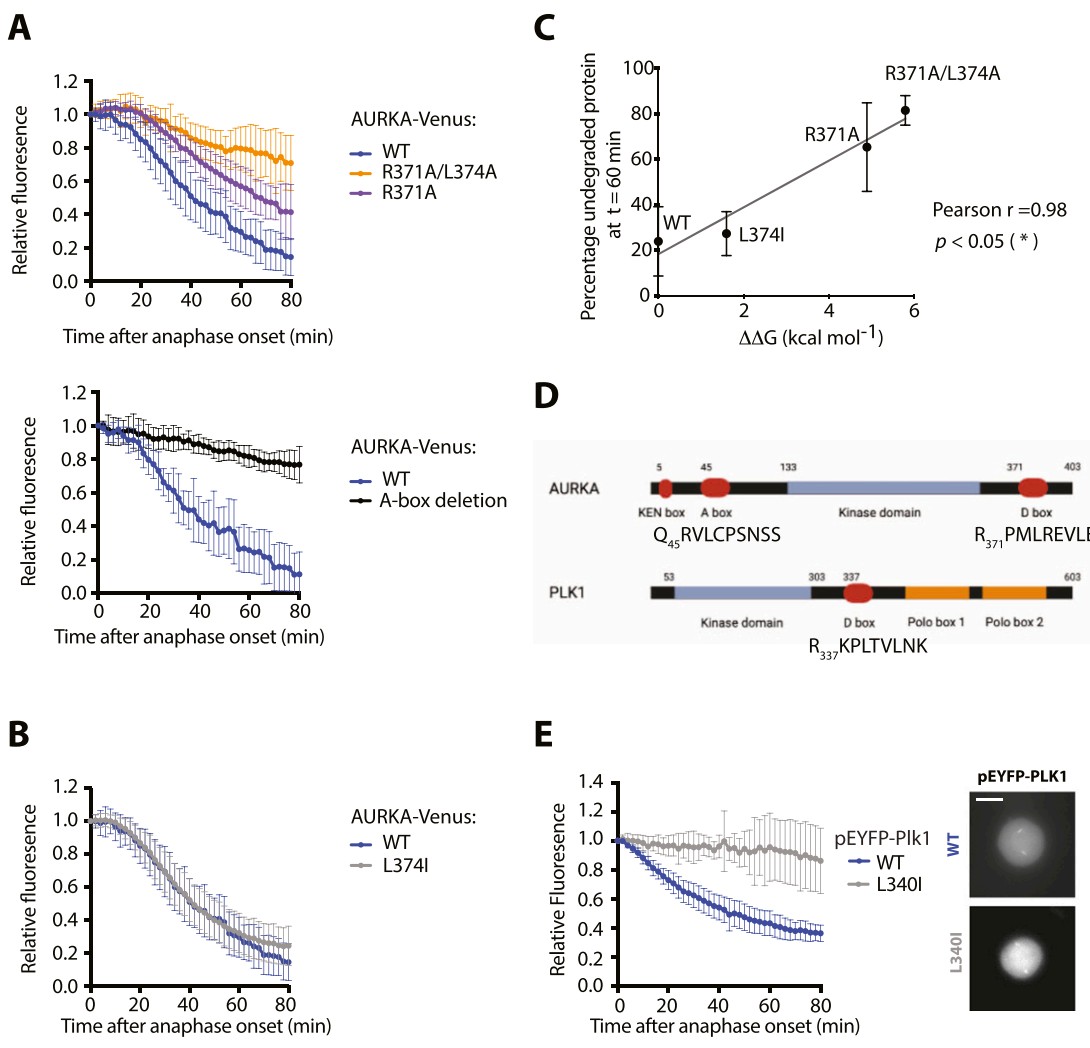

**Figure 2. R$_{371}$xxL motif is not a D-box.**
**(A, B)** In cellulo mitotic degradation assays of AURKA-Venus. Graphs show quantified Venus levels from fluorescence time-lapse imaging of single cells undergoing mitotic exit. Venus levels from individual cells are normalized against the Venus level at anaphase onset and in silico–synchronized so that the mean and SDs can be plotted for each version of AURKA-Venus. **(A)** Mutations predicted to cause disruption of the C-terminus block AURKA degradation during mitotic exit: R371A/L374A (top graph, n ≥ 10 cells, pooled from two experiments) blocks degradation of AURKA-Venus in a similar way to deletion of the N-terminal A-box (bottom graph, n ≥ 6 cells, from a single experiment) (Floyd et al, 2008). **(B)** Conservative substitution L374I has no effect on kinetics of degradation of AURKA-Venus (n ≥ 10 cells, pooled from two experiments). **(C)** Correlation plot for percentage degradation of each version of AURKA-Venus versus predicted ΔΔG of substituted residues in R$_{371}$xxL. **(D)** Schematic of known and proposed degrons in AURKA and Plk1. **(E)** In cellulo mitotic degradation assays of YFP-Plk1 WT and L341I version (n ≥ 10 cells, pooled from two experiments). The L > I substitution at P4 of the R$_{337}$xxL motif abrogates degradation at mitotic exit whilst having no effect on localization of the protein (right-hand panels).

exit degradation by the S51D substitution. Our in silico docking model therefore explains known features of AURKA degradation and assigns the "A-box" degron to the category of phospho-regulated D-boxes alongside securin, CDC6, and KIFC1 (Holt, 2012; Singh et al, 2014). We note that a recent study from the Barford laboratory comparing cryo-EM structures of D-boxes in cyclin A reveals that canonical D-box D1 (RxxL) binds in a different mode to a newly identified non-canonical D-box D2 (VxxL) and that, in in vitro ubiquitination assays, D2 behaves as a stronger D-box degron than D1 (PDB ID:6Q6H; Zhang et al, 2019). Overall, our findings are consistent with the idea that specific degron properties can be encoded in non-canonical D-boxes and that Q$_{45}$RVL is an APC/C$^{FZR1}$-specific D-box.

If all sequences required for APC/C$^{FZR1}$-mediated degradation of AURKA reside in its N-terminal IDR, then why does mutation in the C-terminus stabilize the full-length protein? In this study, we have shown (Fig 1) that the R$_{371}$xxL motif is required for interaction of AURKA with TPX2. Is TPX2-mediated activation of AURKA therefore required for it to be a target of the APC/C? We tested this question by examining mitotic degradation of AURKA S155R-Venus (which interacts only weakly with TPX2; Fig 1E). We found that S155R was degraded with identical kinetics to the WT protein (Fig 5A) and concluded that reduced interaction with TPX2 has no effect upon AURKA degradation. Therefore, we propose that TPX2 per se is not required for AURKA-Venus degradation. Instead, we propose that both interaction of AURKA with TPX2 and interaction of AURKA with

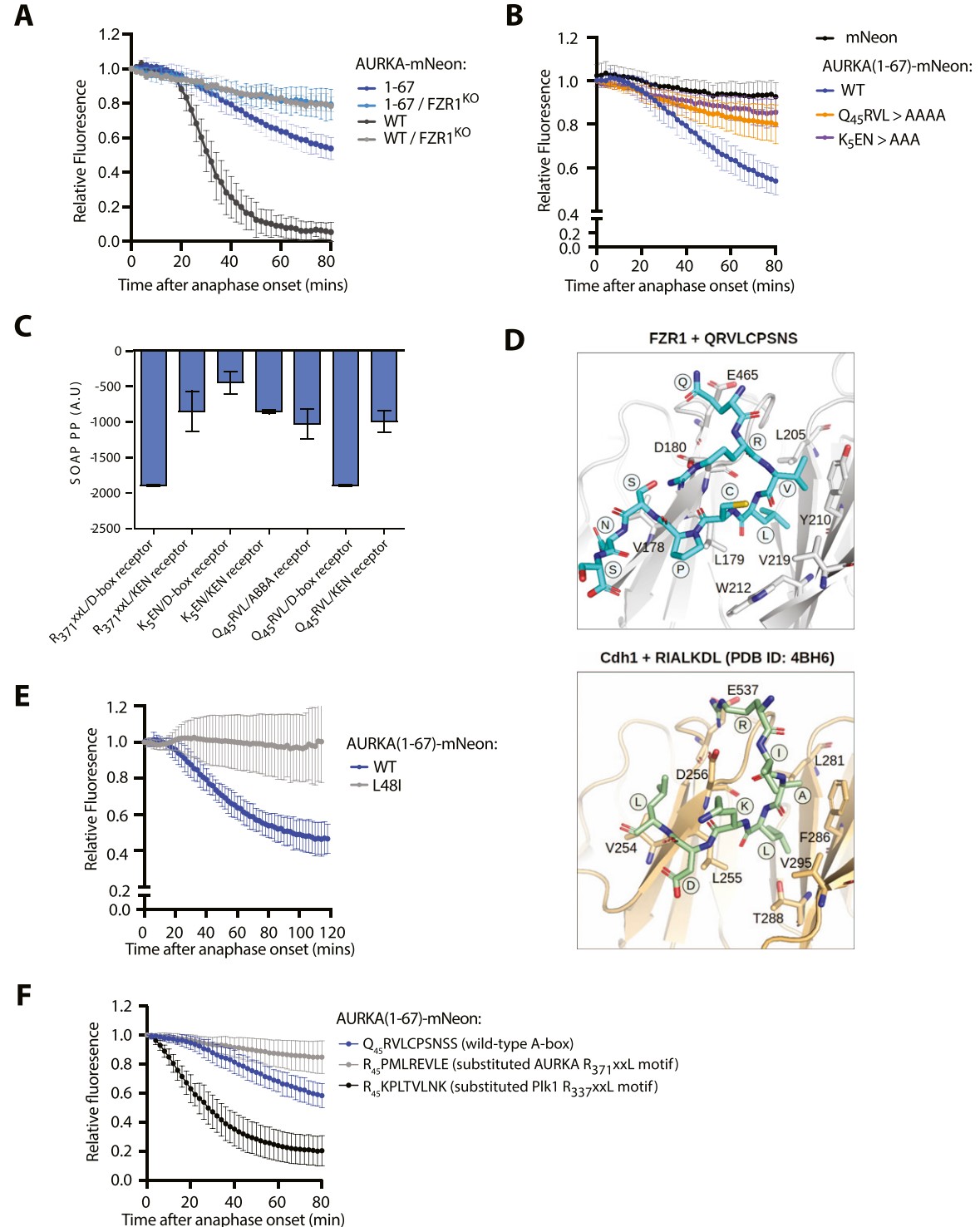

**Figure 3.  Q₄₅RVL motif within the AURKA A-box displays properties of a D-box degron.**

**(A, B)** In cellulo mitotic degradation assays for full-length AURKA-mNeon and AURKA (1-67)-mNeon expressed in U2OS or FZR1[KO] U2OS cells. mNeon levels from individual cells are normalized against the anaphase onset level and in silico–synchronized so that the mean and SDs can be plotted for each protein (n ≥ 20 cells pooled from ≥ 3 experiments). Degradation curves show that (A) residues 1-67 are sufficient for mitotic exit–specific FZR1-dependent degradation and (B) mitotic degradation of AURKA (1-67)-mNeon depends on SLiMs at K₅ and Q₄₅RVL. **(C)** Energetics of in silico docking of proposed AURKA degrons into known binding pockets on FZR1, scored by statistically optimized atomic potential for protein–protein docking, using the FlexPepDock server. **(D)** A-box (QRVLCPSNS) peptide docked to the *H.s.* FZR1 DBR (top panel), modelled upon structure PDB ID:4BH6 that shows *S.c.* Cdh1 bound to the D-box peptide (RIALKD). PDB ID:4BH6 is shown for comparison in the bottom panel (He et al, 2013). **(E)** In cellulo mitotic degradation assays of AURKA-Venus WT and L48I (L > I at P4 of A-box motif, n = 26 cells pooled from two experiments). **(F)** In cellulo mitotic degradation assays of AURKA (1-67)-mNeon in which the Q₄₅RVLCPSNS peptide (the so-called "A-box") is substituted with the *bona fide* D-box from Plk1 (R₃₃₇KPLTVLNK) or with the C-terminal R₃₇₁PMLREVLE motif of AURKA.

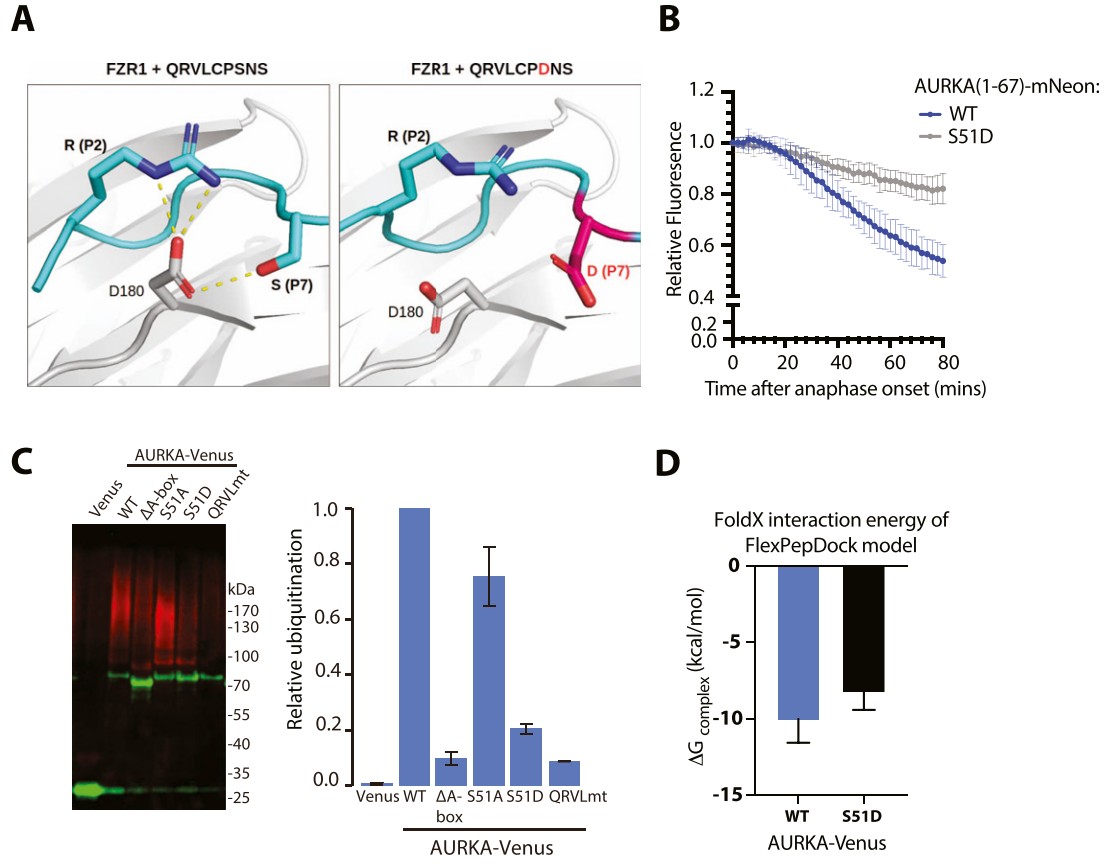

**Figure 4. Modelling of the Q$_{45}$RVL motif at the DBR explains the role of AURKA S51 phosphorylation.**
**(A)** Docking of Q$_{45}$RVLPSNSS peptide on FZR1. Predicted pose for QRVL in the DBR by in silico docking, showing orientation of the P4 leucine and novel contacts afforded at P2 and P7. **(B)** In cellulo mitotic degradation assays of AURKA-Venus WT and S51D. **(C)** Ubiquitination of WT and different versions of AURKA carrying mutations in the A-box; transiently expressed Venus-tagged proteins were purified from U2OS cells synchronized in mitotic exit and blotted for GFP (in green) and ubiquitin conjugates (FK1 antibody, in red). Relative ubiquitination was plotted as the ratio of ubiquitin-conjugated:unmodified protein, normalized against the WT protein; error bars show SDs from three repeats of the experiment. **(D)** Free energy values for WT and S51D peptides computed using FoldX3. In silico docking models were rebuilt using the mutant peptide QRVLCPDNS, models were scored with FoldX3, and the average binding free energies of 10 models for each were plotted. The higher binding energy of the mutant is significant according to a Mann–Whitney test ($P = 0.0147$).

APC/C$^{FZR1}$ depend upon a conformational state blocked by disruption of the R$_{371}$xxL motif in the C-terminal helix of the kinase domain.

Taken together, our results indicate that mutation in R$_{371}$ prevents the active, degradable conformation of AURKA. An autoinhibitory state involving interaction between the kinase domain and the N-terminus of AURKA has previously been described (Zhang et al, 2007; Bai et al, 2014), and studies with a FRET-based conformational sensor confirm that the relative configuration of N- and C-terminus is altered upon activation of the kinase (Bertolin et al, 2016). Because interaction with APC/C$^{FZR1}$ occurs through an N-terminal degron motif, we propose that the Q$_{45}$xxL D-box is "buried" in the inactive conformation, which may be the previously described autoinhibited state mediated through interaction between N- and C-terminal domains (Fig 5B). Intriguingly, a heterotetrameric structure for AURKA-TPX2 captured by the Kern laboratory (Zorba et al, 2014) shows AURKA as a dimer stabilized by a "dimer swap" interaction of R371 residue with E299 in a second AURKA molecule. That is, the usual stabilizing interaction between R371 and E299 can occur in an intermolecular fashion instead of an intramolecular fashion and the authors propose dimerization as a necessary step in autoactivation of the kinase (promoting autophosphorylation, and allowing release of the autoinhibited state and interaction with TPX2). The most recent published studies of AURKA have described redox regulation of AURKA activity through oxidation or inhibitory CoA-lation of a cysteine residue in the kinase activation loop (Byrne et al, 2020; Tsuchiya et al, 2020) also implicated in AURKA dimerization in vitro (Zorba et al, 2014; Tsuchiya et al, 2020), indicating that there is still much to learn of the conformational complexities of AURKA.

Our own findings are consistent with the idea that R371 facilitates an activation step of AURKA and reconciles a large amount of disparate and sometimes contradictory literature on the role of the C-terminal R$_{371}$xxL (historically referred to as the "D-box") in the activity, function, and degradation of AURKA. In summary, we present evidence that the previously assigned "D-box" (R$_{371}$xxL) of AURKA is not a degron and that the previously named "A-box" degron (Q$_{45}$RVL) is the *bona fide* AURKA D-box. Given the timing and dependencies of AURKA degradation, we propose that the A-box should be considered an APC/C$^{FZR1}$-specific D-box.

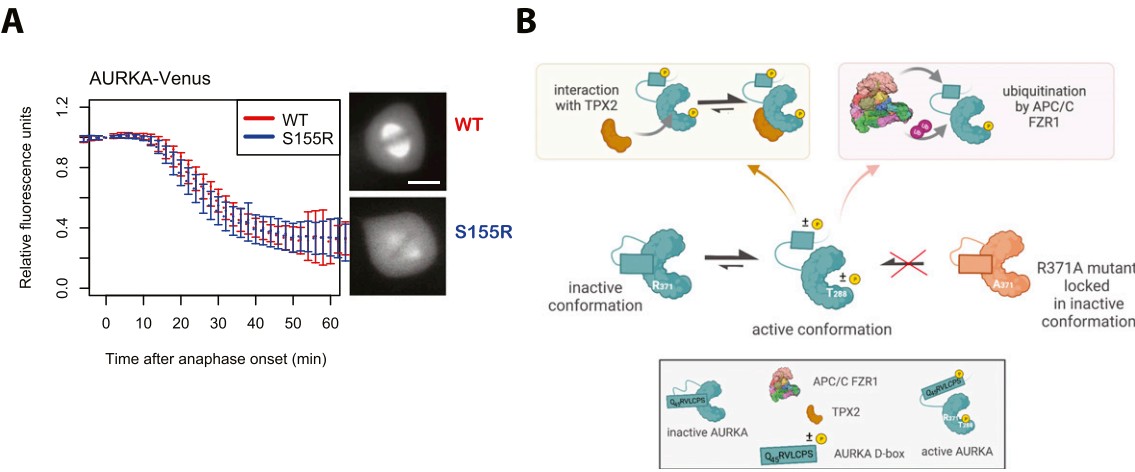

**Figure 5.  R$_{371}$xxL motif plays a critical role in conformational regulation of AURKA.**
**(A)** In cellulo mitotic degradation assays of AURKA-Venus WT and S155R. Venus levels from individual cells are normalized against the anaphase onset level and in silico–synchronized so that the mean and SDs can be plotted for each protein (n ≥ 10 cells, representative of two experiments). **(B)** Schematic proposing that the link between R371 and degradability of AURKA is mediated by a conformational step that simultaneously activates AURKA (leading to phosphorylation on T288) and makes it degradable by the APC/C. The Q$_{45}$RVL motif is "buried" in the autoinhibited state of the WT protein (green) and once released is autoregulated by phosphorylation on S51. The R371A mutant (orange) is unable to undergo the critical conformational step required for both activation and degradation. Schematic created in BioRender.com.

## Materials and Methods

### In silico mutagenesis and docking

Variation in free energy of folding upon point mutations was estimated using the FoldX program (Guerois et al, 2002). Docking was performed using FlexPepDock to return a docked model for each peptide tested. Where indicated, these were scored for comparison using the statistically optimized atomic potential for protein–protein docking assigned to each interaction (Dong et al, 2013). Docking simulations at the DBR were filtered for L4 in the same position as for Acm1 bound to *Saccharomyces cerevisiae* Cdh1 (PDB ID:4BH6). No positional restraints were applied when docking at the KEN/ABBA sites.

### Plasmids

pVenus-AURKA-N1 has been described in previous publications (Min et al, 2013). N-terminal fragments 1-133 and 1-67 of AURKA were cloned into pEGFP-N1, with EGFP substituted by mNeon-HA for later experiments. Point mutagenesis on AURKA was carried out using standard techniques; all cloning details are available upon request.

### Cell culture, synchronization, and transfection

U2OS cells were cultured in DMEM supplemented with 10% FBS, 200 $\mu$M GlutaMAX-1, 100 U/ml penicillin, 100 $\mu$g/ml streptomycin, and 250 ng/ml Fungizone (all from Thermo Fisher Scientific) at 37°C in a humidified atmosphere containing 5% $CO_2$. Cells were transfected using a Neon electroporator (Thermo Fisher Scientific) and seeded onto eight-well microscope slides (Ibidi) for live-cell imaging or onto coverslips for PLAs.

For synchronization in mitosis, cells were treated for 12 h with 10 $\mu$M STLC (Tocris Bioscience) to trigger the spindle assembly

checkpoint. For mitotic exit synchronizations, cells synchronized with STLC were collected by shake-off and released by 70-min treatment with 10 $\mu$M ZM447439, an inhibitor of AURKB that ablates spindle assembly checkpoint arrest.

### Fluorescence microscopy of living cells

A few hours before live-cell imaging of Ibidi slides, the culture medium was changed to L15 supplemented with 10% FBS, antibiotics, and antimycotics. Images were acquired using an automated epifluorescence imaging platform composed of an Olympus IX83 motorized inverted microscope, Spectra-X multi-channel LED widefield illuminator (Lumencor, Inc.), OptoSpin filter wheel (Cairn Research), CoolSnap MYO CCD camera (Photometrics), automated XY stage (ASI), and climate chamber (Digital Pixel), and controlled using Micro-Manager (Edelstein et al, 2014). For mitotic degradation assays, multiple fields containing Venus-, EGFP-, or mNeon-positive prophase and metaphase cells were selected. Images were then acquired every 2 min for 2 h using appropriate filter sets. Fluorescence levels were measured using ImageJ and degradation curves plotted in GraphPad Prism, and in silico–synchronized to anaphase onset for each construct tested.

### IF microscopy and isPLA

U2OS cells were transfected with AURKA-Venus WT or mutant and seeded onto coverslips. After 24 h, cells were fixed with ice-cold methanol (IF) or 4% PFA (isPLA). Cells were processed for IF using primary antibodies against GFP (rabbit polyclonal, ab290; Abcam) and $\beta$-tubulin (mouse mAb, T4026; Sigma-Aldrich) and secondary antibodies anti-mouse Alexa Fluor 488 and anti-rabbit Alexa Fluor 568 (Thermo Fisher Scientific). Cells were processed for isPLA using Duolink In Situ Detection Orange (Sigma-Aldrich) according to the manufacturer's instructions, using primary antibodies against GFP

(mouse mAb, clone #11814460001; Roche) and TPX2 (rabbit poly-clonal antibody; Novus Biological). Epifluorescence images were acquired on the widefield imaging platform described above, as stacks of 500-nm step with 2 × 2 bin, using appropriate filter sets and 40× NA 1.3 oil objective, and exported as maximal intensity projections in ImageJ (http://rsb.info.nih.gov/ij/; National Institutes of Health).

### Immunoblotting

Cells were lysed in 1% Triton X-100, 150 mM NaCl, 10 mM Tris–HCl at pH 7.5, and EDTA-free protease inhibitor cocktail (Roche), and PhosSTOP inhibitor for phosphatase (Sigma-Aldrich). After 30 min on ice, the lysate was centrifuged at 16,000$g$ (4°C) for 10 min. For immunoblotting, an equal amount of protein (20 $\mu$g) was loaded into SDS–PAGE 4–12% precast gradient gels. Proteins were transferred to Immobilon-P or Immobilon-FL membranes using the XCell IITM Blot Module according to the manufacturer's instructions. Membranes were blocked in PBS, 0.1% Tween-20, and 5% BSA and processed for immunoblotting. Primary antibodies for immunoblot were as follows: AURKA (1:1,000; mouse mAb, clone 4/IAK1; BD Transduction Laboratories), phospho-Aurora A (Thr288)/Aurora B (Thr232)/Aurora C (1:1,000; XP rabbit mAb, clone D13A11; Cell Signaling Technology), $\beta$-tubulin (1:2,000; rabbit polyclonal, ab6046; Abcam), and GAPDH (1:400; rabbit mAb, clone #2118; Cell Signaling Technology). Secondary antibodies used were HRP-conjugated, or IRDye 680RD– or 800CW-conjugated at 1:10,000 dilution for quantitative fluorescence measurements on an Odyssey Fc Dual-Mode Imaging System (LI-COR Biosciences).

### Ubiquitination assays

U2OS-bioUb cells transfected with different versions of AURKA-Venus were synchronized as described above and processed for detection of ubiquitin conjugates as previously described (Min et al, 2013).

### Statistical analysis

Data analyses were performed in GraphPad Prism. Results were analysed with a $t$ test, a Mann–Whitney U test (non-parametric), or ordinary one-way ANOVA as indicated in figure legends. Significant results are indicated as $P < 0.05$ (*), $P \leq 0.01$ (**), or $P \leq 0.001$(***). Values are stated as the mean ± SDs.

## Supplementary Information

## Acknowledgements

We thank Norman Davey, Laura Itzhaki, and Rohan Eapen for discussions during the course of this work and many past researchers and students in the laboratory for their contributions to understanding AURKA degradation.

This project was supported by the Royal Society International Exchanges award (IES/R3/170195) to C Lindon and G Guarguaglini, whereas work in C Lindon's laboratory was supported by Cancer Research UK (C3/A10239) and BBSRC (BB/R004137/1). Studentship support is acknowledged by A Abdelbaki from Yousef Jameel Scholarship (Cambridge International Trust), by CN Okoye from Gates Cambridge and Rosetrees Trust, and by C Ascanelli from AstraZeneca UK. A Paiardini and G Guarguaglini are funded by Associazione Italiana Ricerca sul Cancro (AIRC MFAG id. 20447 and 25648 respectively).

## Author Contributions

A Abdelbaki: investigation and writing—review and editing.
C Ascanelli: formal analysis, investigation, methodology, and writing—review and editing.
CN Okoye: formal analysis, investigation, methodology, and writing—review and editing.
HB Akman: formal analysis and investigation.
G Janson: conceptualization, supervision, investigation, and methodology.
M Min: investigation.
C Marcozzi: investigation.
A Hagting: investigation.
R Grant: investigation.
M De Luca: investigation.
IA Asteriti: supervision and methodology.
G Guarguaglini: conceptualization, resources, funding acquisition, project administration, and writing—review and editing.
A Paiardini: conceptualization, resources, supervision, and writing—review and editing.
C Lindon: conceptualization, resources, data curation, formal analysis, supervision, funding acquisition, investigation, methodology, project administration, and writing—original draft, review, and editing.

## Conflict of Interest Statement

The authors declare that they have no conflict of interest.

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
