## [Reviewer comments · Life Science Alliance]

Life Science Alliance

Revisiting degron motifs in human AURKA required for its targeting by APC/C-FZR1

Catherine Lindon, Ahmed Abdelbaki, Camilla Ascanelli, Cynthia Okoye, Hesna Akman, Giacomo Janson, Ming-wei Min, Chiara Marcozzi, Anja Hagting, Rhys Grant, Maria De Luca, Italia Asteriti, Giulia Guarguaglini, and Alessandro Paiardini

DOI: <https://doi.org/10.26508/lsa.202201372>

Corresponding author(s): Catherine Lindon, University of Cambridge

Review Timeline:	Submission Date:	2022-01-16
	Editorial Decision:	2022-02-14
	Revision Received:	2022-10-19
	Editorial Decision:	2022-11-16
	Revision Received:	2022-11-17
	Accepted:	2022-11-18

Scientific Editor: Novella Guidi

Transaction Report:

February 14, 2022

Re: Life Science Alliance manuscript #LSA-2022-01372

Dr. Catherine Lindon
University of Cambridge
Pharmacology
Tennis Court Road
Cambridge CB2 1PD
United Kingdom

Dear Dr. Lindon,

Thank you for submitting your manuscript entitled "Revisiting degron motifs in human AURKA required for its targeting by APC/C-FZR1" to Life Science Alliance. The manuscript was assessed by expert reviewers, whose comments are appended to this letter. We, thus, encourage you to submit a revised version of the manuscript back to LSA that responds to all of the reviewers' points.

Thank you for this interesting contribution to Life Science Alliance. We are looking forward to receiving your revised manuscript.

Sincerely,

B. MANUSCRIPT ORGANIZATION AND FORMATTING:

Reviewer #1 (Comments to the Authors (Required)):

The manuscript from Abdelbaki et al provides a structural and functional characterization of two domains contained in the multifunctional protein AURKA. It is known that the abundance of AURKA peaks at the G2/M checkpoint, and then the kinase is degraded at the end of the mitotic phase/early G1 by the APC/C E3 ubiquitin ligase complex. AURKA possesses multiple motifs - commonly called degrons - recognized by the APC/C complex and targeted for ubiquitylation. The manuscript focuses on two of these motifs, the C-ter D-box and the N-ter A-box.

By combining structural predictions and cell biology approaches, the manuscript provides important advances in our understanding of the mode of regulation of AURKA. In particular, it provides evidence on the fact that the C-ter D-box is not a degron, and the A-box is instead a D-box degron.

This manuscript is certainly interesting, very well written, and it sheds new light on key structural features of AURKA. This could open up new venues for therapeutic possibilities in patients with cancers showing the overexpression of AURKA and where, as the authors state, the actual strategies show poor efficacy. The experiments presented in the manuscript are convincing, and the manuscript style is well adapted to the broad readership of the journal. Globally, this is a very good manuscript and it deserves publication in Life Science Alliance.

Nevertheless, I have few points that I believe should be addressed before the manuscript is published in its final form.

MAJOR POINTS

1. The R317A mutation of AURKA is presented as "weakly present on the microtubules of the mitotic spindle". I would recommend to avoid definitions as "weak" or "strong" without providing quantifications, as these words can be prone to interpretation. Representative micrographs without an accompanying quantification should be avoided. Indeed, the R317A variant has a diffused cytoplasmic distribution, yet it appears that it can localize to the spindle as the WT construct. To avoid ambiguity, the variants presented in the manuscript (Fig. 1C) should be characterized more thoroughly, and this by (a) quantifying the proportion of cells with mitotic spindles when each of these variants is expressed, (b) their capacity to colocalize with a spindle marker, and (c) the cytoplasmic vs spindle fluorescence ratio for each of the mutants.
2. The isPLA assay is a visual way to demonstrate the interaction between two proteins. However, it is unclear to me whether the R317A/L374A double mutant really does not show any interaction. If no interaction exists, the PLA assay should not yield any or very little fluorescent dots (as in Supplementary Fig. 2A and B). However, the micrographs and the quantification in Fig. 1D show that there is a certain degree of interaction between AURKA R317A/L374A double mutant and TPX2, which is normalized around 1. Therefore, it would be more prudent to tone down the conclusion of "no interaction". Besides, I am surprised to see that the S155R mutant lowers, but it does not abolish the interaction with TPX2. The degree of interaction of AURKA S155R with TPX2 is similar to the one between AURKA L374I and TPX2, yet the L374I mutant shows little differences with the WT protein in terms of structural properties and functional consequences on the spindle. Do the mutants show differential interactions among them, or somehow there is a "full interaction" for the WT and a "lowered" interaction for the mutants, regardless of the nature of the mutant? Last, the isPLA staining is reminiscent of the spindle. Does that mean that the interaction takes place on the spindle for all the mutants? If so, how do the authors reconcile the fact that some of these mutants as the R317A/L374A double mutant or the S155R do not localize to the spindle? Where does the interaction take place?
3. The AURKA-mNeon 1-67 mutant is only partially degraded after anaphase onset. According to the experiment shown, there is between 70% (Fig. 3A), 60% (Fig. 3B) and 50% (Fig. 3E) remaining on the spindle. This argues against the fact that this domain "recapitulates the known regulation of degradation of the full-length protein". I would suggest discussing this partial degradation more in depth, and to tone down this last statement.
4. To conclude on the capacity of different degron motifs/residues to be targeted by ubiquitylation, the results obtained with the FK1 antibody should be correlated with assays to evaluate the capacity of these constructs to be targeted by K48-specific ubiquitylation. With the exception of the QRVL mutant, which shows a very faint FK1-positive smear, the others show more significant ubiquitylation-related patterns. Surprisingly, even the S51D does so, although it would be expected to correspond to a zero-ubiquitylation condition. It would be important to exclude that other modes of ubiquitylation are taken up by these

constructs, and that they still recapitulate a proteasome-mediated degradation. In addition, formal proof that the R371xxL domain is not ubiquitylated is still missing, and should be provided to corroborate that this domain is indeed not a D-box.

5. I really enjoyed the kinetics shown in Fig. 5. I believe that putting this experiment as the last piece of evidence is a nice way to finish the story. However, the conclusion of the manuscript cannot be so strict. This is in light of the interactions shown in Fig. 1D between AURKA S155R and TPX2 which are lowered, but not abolished. Therefore, one cannot completely ascertain that there is no requirement of a functional interaction with TPX2 for AURKA to be degraded. Can the S155R approach be corroborated by degradation analyses in the absence of endogenous TPX2, since there is still the endogenous one remaining in the analyses shown in Fig. 5? In addition, the R371A simple mutant partially interacts with TPX2, and it is partially degraded after anaphase onset. If the interaction with TPX2 ends up being required for AURKA degradation after anaphase, would a R371A mutation in the presence of a S155R mutation totally abolish the degradation of the kinase after anaphase?

6. Is there a link between the capacity of full-length AURKA to be ubiquitylated, to interact with TPX2 and to be degraded after anaphase onset? In other terms, is the interaction with TPX2 co-occurring with ubiquitylation events, or a prerequisite for the kinase to be targeted for ubiquitylation?

MINOR POINTS

1. The western blotting assay presented in Supplementary Fig. 1C is of poor quality. The background has been lowered in order to lower the saturation of the AURKA-specific bands, which are often irregular and smeared. The tubulin-specific bands are also irregular for the R371A mutant. Is this simply a loading problem, or is it a real biological event of the mutant on tubulin? This blot should be repeated to dissipate the doubt.
2. In isPLA graphs, are the independent dots issued from the same experiment or are they issued from the replicates and plotted together? If so, they should not be considered as independent n for statistics
3. The legend of Supplementary Fig. 2 is partially cut.
4. The graph in Fig. 2E indicates "L341I", and it should be "L340I" instead.

Reviewer #2 (Comments to the Authors (Required)):

1. During mitosis, the localisation and catalytic activity of Aurora-A kinase is controlled through protein-protein interactions (most notably, TPX2) and then, in late mitosis, Aurora-A is ubiquitinated by APC/C and targeted for degradation. One puzzling aspect of the current model is the motifs in Aurora-A that are required. A supposedly key motif (D-box) is buried in the 3D structure of the kinase, and is therefore unlikely to play a role. This is a 20-year old problem that everyone has ignored. This paper takes this long-standing question on, and does so rigorously, elegantly and convincingly. The answer is in 2 parts. First, another motif (previously described as an A-box is actually the D-box). Second, the previously-described D-box motif is important structurally, and mutations here result in inactive protein that do not bind TPX2 and do not localise properly. I very much appreciated the approach taken by the investigators, using a combination of computational analysis of structures and cell-based assays.

The paper raises a new and interesting question - why would a mutation in the C-lobe of Aurora-A affect binding to: (i) TPX2, which binds primarily to the N-lobe; (ii) and APC/C, which binds to the extended N-terminal region? The authors put forward a plausible explanation, making good use of the literature. Although additional work would be required to test this model, I do not think it is needed for this publication.

2&3 I have read this manuscript thoroughly, and I cannot find any issues to be addressed.

Reviewer #3 (Comments to the Authors (Required)):

Aurora A kinase (AURKA) is a mitotic kinase with critical roles in centrosome maturation and mitotic spindle assembly. AURKA activity is regulated by proteolysis mediated by the Anaphase Promoting Complex/Cyclosome (APC/C FZR1) during mitotic exit and G1 phase. Previous work reported that AURKA degradation requires an atypical N-terminal degron called the A box and a canonical D-box degron (RxxL) in the C-terminal part of the protein. Notably, the D-box is located in the highly structured kinase domain and is not accessible, raising questions about the exact contribution of the D-box for physiological AURKA degradation. In this manuscript, Abdelbaki et al. convincingly demonstrate that the C-terminal D-box is not a degron, but rather a structural element of AURKA, required for its normal function. By contrast, they show that the previously named A-box located in the N-terminal part of the protein is a bona fide D-box.

Overall the experiments are carefully done and generally support the main conclusions with a few exceptions listed below.

Major points:

1- At the bottom of page 4 and also on other occasions, the authors write: "We concluded that perturbed folding of the C-terminal region...etc". However, the authors did not directly demonstrate that mutation of the C-terminal D-box perturbs the folding of AURKA, even though it is an interesting and plausible hypothesis.

2- In Figure 1 the authors should clearly state where is the real structure and where are the models.

3- PLA is not exactly a direct measure of protein-protein interaction. To be able to conclude that AURKA substitutions in the D-box affects Tpx2 binding, the authors should measure affinities with purified proteins.

Minor points:

3- The authors should present some time-lapse of their movies in support of the graphs presented in the main figures.

4- Figure 1D lacks scale bars.

Reviewer #1 (Comments to the Authors (Required)):

The manuscript from Abdelbaki et al provides a structural and functional characterization of two domains contained in the multifunctional protein AURKA. It is known that the abundance of AURKA peaks at the G2/M checkpoint, and then the kinase is degraded at the end of the mitotic phase/early G1 by the APC/C E3 ubiquitin ligase complex. AURKA possesses multiple motifs - commonly called degrons - recognized by the APC/C complex and targeted for ubiquitylation. The manuscript focuses on two of these motifs, the C-ter D-box and the N-ter A-box.

By combining structural predictions and cell biology approaches, the manuscript provides important advances in our understanding of the mode of regulation of AURKA. In particular, it provides evidence on the fact that the C-ter D-box is not a degron, and the A-box is instead a D-box degron.

This manuscript is certainly interesting, very well written, and it sheds new light on key structural features of AURKA. This could open up new venues for therapeutic possibilities in patients with cancers showing the overexpression of AURKA and where, as the authors state, the actual strategies show poor efficacy. The experiments presented in the manuscript are convincing, and the manuscript style is well adapted to the broad readership of the journal. Globally, this is a very good manuscript and it deserves publication in Life Science Alliance.

Nevertheless, I have few points that I believe should be addressed before the manuscript is published in its final form.

MAJOR POINTS

1. The R317A mutation of AURKA is presented as "weakly present on the microtubules of the mitotic spindle". I would recommend to avoid definitions as "weak" or "strong" without providing quantifications, as these words can be prone to interpretation. Representative micrographs without an accompanying quantification should be avoided. Indeed, the R371A variant has a diffused cytoplasmic distribution, yet it appears that it can localize to the spindle as the WT construct. To avoid ambiguity, the variants presented in the manuscript (Fig. 1C) should be characterized more thoroughly, and this by (a) quantifying the proportion of cells with mitotic spindles when each of these variants is expressed, (b) their capacity to colocalize with a spindle marker, and (c) the cytoplasmic vs spindle fluorescence ratio for each of the mutants.

We agree that qualitative use of the word 'weak' lacks rigour. We had not thought it important to quantify the observation of 'weak' localization of R317A to spindle since we have quantified the interaction with TPX2 which has been shown by others to determine the localization of AURKA on the spindle. However these data can readily be presented - as suggested - as spindle vs cytoplasmic fluorescence ratios (suggestion c). Therefore we carried out **new IF experiments** that would allow us to measure the spindle:cytoplasmic ratio of different versions of AURKA-Venus, whilst simultaneously staining for tubulin to confirm the presence of the mitotic spindle in cells lacking spindle localization of AURKA-Venus variants. We have now added these data to Figure 1 (**Figure 1D**), whilst moving one of the other figure panels to Supplementary data (new **Figure S2**).

There is no evidence that R371A or any other variant produces a mitotic arrest phenotype (endogenous AURKA is present) - so we don't believe that the suggested quantification of the proportion of mitotic spindles (suggestion a) is a useful addition to our characterization of these mutants. Moreover the role of AURKA-TPX2 interaction has been characterized in great detail by others (notably Bird & Hyman 2008 study). We do agree that normalizing the localization of AURKA variants versus another spindle marker (suggestion b) would be a nice parameter for further characterization of the mutants. However we could not get useful data from our IF experiments

because of considerable variations in levels of expression of the transfected AURKA construct. We are confident that the spindle:cytoplasmic localization differences – which occur in presence of the mitotic spindle – are highly significant (and supported by additional data in the figure).

2. The isPLA assay is a visual way to demonstrate the interaction between two proteins. However, it is unclear to me whether the R371A/L374A double mutant really does not show any interaction. If no interaction exists, the PLA assay should not yield any or very little fluorescent dots (as in Supplementary Fig. 2A and B). However, the micrographs and the quantification in Fig. 1D show that there is a certain degree of interaction between AURKA R371A/L374A double mutant and TPX2, which is normalized around 1. Therefore, it would be more prudent to tone down the conclusion of "no interaction". Besides, I am surprised to see that the S155R mutant lowers, but it does not abolish the interaction with TPX2. The degree of interaction of AURKA S155R with TPX2 is similar to the one between AURKA L374I and TPX2, yet the L374I mutant shows little differences with the WT protein in terms of structural properties and functional consequences on the spindle. Do the mutants show differential interactions among them, or somehow there is a "full interaction" for the WT and a "lowered" interaction for the mutants, regardless of the nature of the mutant? Last, the isPLA staining is reminiscent of the spindle. Does that mean that the interaction takes place on the spindle for all the mutants? If so, how do the authors reconcile the fact that some of these mutants as the R371A/L374A double mutant or the S155R do not localize to the spindle? Where does the interaction take place?

We thank the reviewer for careful critique of the isPLA data. We note that in the control experiment (previously Supplementary S2, now Supplementary S1), there is only one antibody present in each condition, so this cannot be compared with the R371A/R374A to conclude that there is signal in the latter sample. However given that isPLA is inherently noisy, we agree that it would be safer to tone down our conclusions, and have replaced the text in question with:

"As expected we found that the R371A/L374A version of AURKA, which does not localize to the mitotic spindle, showed reduced interaction with TPX2".

We were also disappointed with S155R result. One possible explanation is that – if the heterotetrameric structure [TPX2-AURKA]₂ proposed by [Zorba et al. 2014] exists in cells, then endogenous AURKA might support recruitment of the exogenous mutant AURKA into complex with TPX2.

The very high sensitivity of the assay means that signal is amplified/generated even when there is limited colocalization – and since all of the endogenous TPX2 in the mitotic cell is localized to the spindle, this is where any signal is detected by eye. However the signal is quantified over the whole cell.

3. The AURKA-mNeon 1-67 mutant is only partially degraded after anaphase onset. According to the experiment shown, there is between 70% (Fig. 3A), 60% (Fig. 3B) and 50% (Fig. 3E) remaining on the spindle. This argues against the fact that this domain "recapitulates the known regulation of degradation of the full-length protein". I would suggest discussing this partial degradation more in depth, and to tone down this last statement.

Again, we agree with this. Our statement was intended to be qualitative (about parameters rather than extent of degradation) but could be misleading. We've rewritten it as follows:

"Therefore AURKA (1-67) recapitulates known features of AURKA degradation, although the reduced efficiency of degradation compared to full-length protein leaves open the possibility that additional degrons exist."

4. To conclude on the capacity of different degron motifs/residues to be targeted by ubiquitylation, the results obtained with the FK1 antibody should be correlated with assays to evaluate the capacity of these constructs to be targeted by K48-specific ubiquitylation. With the exception of the QRVL mutant, which shows a very faint FK1-positive smear, the others show more significant

ubiquitylation-related patterns. Surprisingly, even the S51D does so, although it would be expected to correspond to a zero-ubiquitylation condition. It would be important to exclude that other modes of ubiquitylation are taken up by these constructs, and that they still recapitulate a proteasome-mediated degradation. In addition, formal proof that the R371xxL domain is not ubiquitylated is still missing, and should be provided to corroborate that this domain is indeed not a D-box.

We think it is not surprising that S51D shows more ubiquitination than QRVL. Our modelling in Figure 4D shows there is still predicted binding of S51D, but with lower affinity. We've discussed in a previous article (Lindon 2016) that the sigmoidal degradation curve of AURKA implies a non-processive ubiquitination process coupled to a minimum threshold of ubiquitination required for degradation. So the correlation between ubiquitination and degradation is not linear.

We're not clear what additional cell-based experiments could constitute formal proof that R371xxL is not a D-box. The R371A/L374A mutant (in context of full-length protein) is likely not ubiquitinated, but this fact would neither confirm nor refute the D-box status (in the same way that non-degradation of this mutant does not prove that the motif is a D-box).

It would certainly be a good experiment to show – for example – that there were ubiquitination of a lysine residue downstream of the Q45RVL motif and no ubiquitination of any lysine residue downstream of the R371xxL motif, but these would be findings consistent with our hypothesis, and not direct proof. We have previously shown that lysines at both N- and C-terminus of AURKA are ubiquitinated in mitotic extracts (Min et al. 2013).

However in order to strengthen our conclusions **we have added new data** that we believe adds compelling evidence that R371xxL is not a D-box: We have tested degradation of a construct where R371xxL is embedded in an ectopic IDR sequence context (the N-terminal IDR) known to support D-box function (ie we have replaced the N-terminal 'A box' of AURKA (Q45xxL, degraded) with the C-terminal R371xxL of AURKA (not degraded) or R337xxL of Plk1 (degraded). These new data are now in **Figure 3F**.

5. I really enjoyed the kinetics shown in Fig. 5. I believe that putting this experiment as the last piece of evidence is a nice way to finish the story. However, the conclusion of the manuscript cannot be so strict. This is in light of the interactions shown in Fig. 1D between AURKA S155R and TPX2 which are lowered, but not abolished. Therefore, one cannot completely ascertain that there is no requirement of a functional interaction with TPX2 for AURKA to be degraded. Can the S155R approach be corroborated by degradation analyses in the absence of endogenous TPX2, since there is still the endogenous one remaining in the analyses shown in Fig. 5? In addition, the R371A simple mutant partially interacts with TPX2, and it is partially degraded after anaphase onset. If the interaction with TPX2 ends up being required for AURKA degradation after anaphase, would a R371A mutation in the presence of a S155R mutation totally abolish the degradation of the kinase after anaphase?

With regard to the comment that one cannot completely ascertain that there is no requirement of functional interaction with TPX2 for degradation: We think it a reasonable explanation that if functional interaction with TPX2 were a requirement for degradation, then we should see some effect of S155R mutation even if the effect of S155R on TPX2 interaction is only partial.

However as a way of reducing the certainty of our stated conclusion, we have moved the idea that TPX2 interaction is not required for AURKA degradation into our discussion of what we propose, rather than what we conclude, as follows:

"We found that S155R was degraded with identical kinetics to the wild-type protein (Figure 5A) and concluded that reduced interaction with TPX2 has no effect upon AURKA degradation. Therefore we propose that TPX2 per se is not required for AURKA-Venus degradation. Instead, we propose that interaction of AURKA with TPX2 and with APC/C-FZR1 both depend upon a conformational state blocked by disruption of the R₃₇₁xxL motif in the C-terminal helix of the kinase domain."

In the past we have carried out the TPX2 siRNA experiment suggested. We found the results of this experiment less clear cut than the S155R mutant experiment, because TPX2 siRNA causes a strong cell cycle arrest in prometaphase, when there is no AURKA degradation. Therefore although in this experiment AURKA degradation looks normal in cells that enter anaphase, we have to assume that TPX2 knockdown is incomplete in these cells.

6. Is there a link between the capacity of full-length AURKA to be ubiquitinated, to interact with TPX2 and to be degraded after anaphase onset? In other terms, is the interaction with TPX2 co-occurring with ubiquitination events, or a prerequisite for the kinase to be targeted for ubiquitination?

There is a link, in the sense that we believe all of these events depend on the same active conformation of AURKA.

We have spent a lot of time trying to figure out the link between these events that can explain our data. In a previous publication (Abdelbaki 2020, J Cell Science) we showed that inactivation of AURKA in anaphase occurs independently of its ubiquitination and degradation and may in fact depend upon ubiquitination or degradation of TPX2 by APC/C-Cdc20, with targeting of AURKA by APC/C-FZR1 occurring subsequently.

MINOR POINTS

1. The western blotting assay presented in Supplementary Fig. 1C is of poor quality. The background has been lowered in order to lower the saturation of the AURKA-specific bands, which are often irregular and smeared. The tubulin-specific bands are also irregular for the R371A mutant. Is this simply a loading problem, or is it a real biological event of the mutant on tubulin? This blot should be repeated to dissipate the doubt.

We agree this was not a good quality blot and have removed it from the Supplementary data. The purpose of the experiment was to show that removal of endogenous AURKA did not promote relocalization of mutant versions of exogenous to the mitotic spindle (we wanted to show this because the levels of expression of the exogenous proteins varies over more than an order of magnitude between cells, and are different between constructs, therefore variably competed by endogenous protein). However, the spindle:cytoplasmic ratios we measured in the new IF experiment now shown as **Figure 1D** were remarkably consistent between cells over a wide range of [AURKA-Venus] levels, supporting the idea that competition with endogenous AURKA has little effect on the extent of spindle localization of the mutants.

2. In isPLA graphs, are the independent dots issued from the same experiment or are they issued from the replicates and plotted together? If so, they should not be considered as independent n for statistics

Data have been normalized to the mean of the WT values in each experiment and then pooled.

3. The legend of Supplementary Fig. 2 is partially cut.

This has now been fixed (and the figure is now Supplementary Fig. 1)

4. The graph in Fig. 2E indicates "L341I", and it should be "L340I" instead.

Thanks to the reviewer for spotting this error - now fixed

Reviewer #2 (Comments to the Authors (Required)):

1. During mitosis, the localisation and catalytic activity of Aurora-A kinase is controlled through protein-protein interactions (most notably, TPX2) and then, in late mitosis, Aurora-A is ubiquitinated by APC/C and targeted for degradation. One puzzling aspect of the current model is the motifs in

Aurora-A that are required. A supposedly key motif (D-box) is buried in the 3D structure of the kinase, and is therefore unlikely to play a role. This is a 20-year old problem that everyone has ignored. This paper takes this long-standing question on, and does so rigorously, elegantly and convincingly. The answer is in 2 parts. First, another motif (previously described as an A-box is actually the D-box). Second, the previously-described D-box motif is important structurally, and mutations here result in inactive protein that do not bind TPX2 and do not localise properly. I very much appreciated the approach taken by the investigators, using a combination of computational analysis of structures and cell-based assays.

The paper raises a new and interesting question - why would a mutation in the C-lobe of Aurora-A affect binding to: (i) TPX2, which binds primarily to the N-lobe; (ii) and APC/C, which binds to the extended N-terminal region? The authors put forward a plausible explanation, making good use of the literature. Although additional work would be required to test this model, I do not think it is needed for this publication.

2&3 I have read this manuscript thoroughly, and I cannot find any issues to be addressed. We thank this reviewer for thoroughly reading the manuscript and for the succinct summary of our findings.

Reviewer #3 (Comments to the Authors (Required)):

Aurora A kinase (AURKA) is a mitotic kinase with critical roles in centrosome maturation and mitotic spindle assembly. AURKA activity is regulated by proteolysis mediated by the Anaphase Promoting Complex/Cyclosome (APC/C FZR1) during mitotic exit and G1 phase. Previous work reported that AURKA degradation requires an atypical N-terminal degron called the A box and a canonical D-box degron (RxxL) in the C-terminal part of the protein. Notably, the D-box is located in the highly structured kinase domain and is not accessible, raising questions about the exact contribution of the D-box for physiological AURKA degradation.

In this manuscript, Abdelbaki et al. convincingly demonstrate that the C-terminal D-box is not a degron, but rather a structural element of AURKA, required for its normal function. By contrast, they show that the previously named A-box located in the N-terminal part of the protein is a bona fide D-box.

Overall the experiments are carefully done and generally support the main conclusions with a few exceptions listed below.

Major points:

1- At the bottom of page 4 and also on other occasions, the authors write: "We concluded that perturbed folding of the C-terminal region...etc". However, the authors did not directly demonstrate that mutation of the C-terminal D-box perturbs the folding of AURKA, even though it is an interesting and plausible hypothesis.

It's not completely clear to us how the reviewer wishes us to resolve this point. We have used 'folding' as shorthand for the 'folded conformation' of AURKA – ie referring to the structure of the protein rather than to the dynamic process of folding. However we have replaced references to 'perturbed folding' as follows:

"These results suggested that a perturbed structure could be responsible for both the deficiency in interaction with TPX2 and deficiency in mitotic degradation"

"We concluded that perturbed conformation of the C-terminal region of its kinase domain may prevent AURKA folding and function required for interaction with TPX2."

2- In Figure 1 the authors should clearly state where is the real structure and where are the models. Figure 1A legend has been amended to include PDB number of structure used (PDB 1mq4). Figure 1B legend now reads “*B interactions shown in A are lost in predicted structures modelled for R371A and L374A substitutions.*”

3- PLA is not exactly a direct measure of protein-protein interaction. To be able to conclude that AURKA substitutions in the D-box affects Tpx2 binding, the authors should measure affinities with purified proteins.

We don't believe that the extensive *in vitro* experimentation proposed would be appropriate for this study. These proteins have been shown to interact directly in dozens of publications, and our experiments focus on the interaction observed in the cellular context, which likely depends on parameters that might not be fully reproducible in *in vitro* assays. For example, most of the characterization of AURKA-TPX2 interaction *in vitro* has been achieved using fragments of TPX2 rather than full-length protein, which is difficult to work with.

Minor points:

3- The authors should present some time-lapse of their movies in support of the graphs presented in the main figures.

Movies frames from degradation assays of full-length AURKA wt, R371/L374A and L374I (Figure 2A), Plk1 wt and L341I (Figure 2B) and AURKA (1-67) constructs (Figure 3) have been included in a new Supplementary Figure (now Supplementary Figure S3). Please note AURKA 1-67 fragment is delocalized throughout the cell as it does not interact with any known binding partner.

4- Figure 1D lacks scale bars.

Scale bar has been added to the Figure (now Figure 1E).

November 16, 2022

RE: Life Science Alliance Manuscript #LSA-2022-01372R

Dr. Catherine Lindon
University of Cambridge
Pharmacology
Tennis Court Road
Cambridge CB2 1PD
United Kingdom

Dear Dr. Lindon,

Thank you for submitting your revised manuscript entitled "Revisiting degranulation motifs in human AURKA required for its targeting by APC/C-FZR1". We would be happy to publish your paper in Life Science Alliance pending final revisions necessary to meet our formatting guidelines.

- please use the [10 author names, et al.] format in your references (i.e. limit the author names to the first 10)
- please add your supplementary figure legends to the main manuscript text
- the Data Availability section is blank. Please fill in this section or remove it if you do not have any large scale datasets that was deposited.

Figure Check:

- missing scale bars for Figure 5A; Figure S1; Figure S3B

A. FINAL FILES:

B. MANUSCRIPT ORGANIZATION AND FORMATTING:

Sincerely,

Reviewer #1 (Comments to the Authors (Required)):

I would like to thank the authors for taking the time to answer my questions, explaining their rationale and adding further data supporting their findings in this revised version. Congratulations on this very good manuscript! I'm looking forward to seeing it published in its final version.

Reviewer #3 (Comments to the Authors (Required)):

I apologize for the delay in sending my comments on the revised version of the manuscript by Abdelbaki et al. I have now had the opportunity to read in detail the revised version of the manuscript and the response to the reviewers' comments. The authors have responded satisfactorily to my comments.

November 18, 2022

RE: Life Science Alliance Manuscript #LSA-2022-01372RR

Dr. Catherine Lindon
University of Cambridge
Pharmacology
Tennis Court Road
Cambridge CB2 1PD
United Kingdom

Dear Dr. Lindon,

Thank you for submitting your Research Article entitled "Revisiting degron motifs in human AURKA required for its targeting by APC/C-FZR1". It is a pleasure to let you know that your manuscript is now accepted for publication in Life Science Alliance. Congratulations on this interesting work.

DISTRIBUTION OF MATERIALS:

Again, congratulations on a very nice paper. I hope you found the review process to be constructive and are pleased with how the manuscript was handled editorially. We look forward to future exciting submissions from your lab.

Sincerely,
